# Metabolic Dysfunction-Associated Steatotic Liver Disease Shapes a Distinct Semaphorin–Cytokine Immune Signature in Severe Community-Acquired Pneumonia

**DOI:** 10.3390/ijms26168095

**Published:** 2025-08-21

**Authors:** Branimir Gjurašin, Leona Radmanić Matotek, Lara Šamadan Marković, Neven Papić

**Affiliations:** 1Department for Intensive Care, University Hospital for Infectious Diseases “Dr. Fran Mihaljević”, 10000 Zagreb, Croatia; bgjurasin@bfm.hr; 2Department for Infectious Diseases, School of Medicine, University of Zagreb, 10000 Zagreb, Croatia; 3Department for Immunological and Molecular Diagnostics, University Hospital for Infectious Diseases “Dr. Fran Mihaljević”, 10000 Zagreb, Croatia; lradmanic@bfm.hr; 4Croatian Institute of Public Health, 10000 Zagreb, Croatia; samadanlara@gmail.com; 5Department for Viral Hepatitis, University Hospital for Infectious Diseases “Dr. Fran Mihaljević”, 10000 Zagreb, Croatia

**Keywords:** pneumonia, MASLD, NAFLD, liver steatosis, semaphorins, cytokine, immune response, SEMA7A, CXCL10, IL-10

## Abstract

Metabolic dysfunction-associated steatotic liver disease (MASLD) is increasingly recognized as a modulator of infection severity, yet its impact on the immune response in severe community-acquired pneumonia (sCAP) remains poorly understood. In this prospective cohort study of 108 adults with sCAP, we evaluated the prevalence and prognostic impact of MASLD and performed pathogen-stratified immune profiling of cytokines and semaphorins on hospital days 1 and 5. MASLD was present in 50% of patients and independently associated with early respiratory failure (OR 3.8) and vasopressor-dependent shock (OR 4.0), despite similar sCAP severity at baseline. MASLD patients exhibited distinct immune profiles, including elevated baseline serum levels of SEMA3A, SEMA7A, IL-2, IL-10, IL-17A, CXCL10, and TGF-β1, and reduced SEMA5A. By day 5, the MASLD group exhibited a greater decline in pro-inflammatory mediators compared to non-MASLD patients but failed to upregulate reparative mediators such as SEMA4D and TGF-β1, unlike the non-MASLD group. These kinetics may suggest a maladaptive immune response in MASLD, potentially consistent with early immune exhaustion. Immunokinetic patterns were pathogen-specific, including transient increase in IL-17A and IL-10 in Legionella and Mycoplasma infections, and CXCL10, IL-2, IL-17A, TGF-β1 and IL-10 in influenza. Serum IL-10, CXCL10, SEMA3F, SEMA4D and SEMA7A correlated with organ failure and sCAP complications. These findings underscore the clinical importance of the lung–liver axis and suggest that semaphorins could serve as valuable prognostic biomarkers for identifying high-risk patients.

## 1. Introduction

Severe community-acquired pneumonia (sCAP) is the leading cause of ICU admission, sepsis, and acute respiratory distress syndrome (ARDS), with 30-day mortality exceeding 10% and reaching up to 50% in ICU-treated patients [1,2,3]. Nearly one-quarter of hospitalized CAP cases require intensive care [4], and survivors face long-term risks, including cardiovascular complications [5]. Despite advances in care, sCAP mortality has remained largely unchanged since the antibiotic era.

The host immune response in sCAP is highly heterogeneous and complex, shaped by numerous underexplored host factors. To date, immunomodulatory therapies for sCAP are largely ineffective, except for corticosteroids in narrowly defined indications [6]. A deeper understanding of immunopathogenesis and the discovery of novel sCAP biomarkers are essential for more precise prognostication, more effective immunomodulation, and better clinical outcomes.

Metabolic dysfunction-associated steatotic liver disease (MASLD) is the most common liver disorder, affecting nearly one-third of the adult population [7]. While components of metabolic syndrome (MetS) are well known to increase pneumonia risk and worsen outcomes [8,9,10], a similar association with MASLD has only recently been proposed. Most studies focused on MASLD and COVID-19, with only three studies addressing pneumonia of other etiologies. A recent population study reported a two-fold increased risk of respiratory infection-related hospitalization [11], while another showed higher 30-day mortality in CAP patients with MASLD (17% vs 6%), although sCAP patients were not included [12]. In our retrospective study of ICU-treated sCAP patients, MASLD was highly prevalent (58%) and independently associated with both severe ARDS (69% vs 43%) and increased mortality (50% vs 21%) [13].

The mechanisms linking MASLD with worse sCAP outcomes remain unclear—they may involve immune dysregulation driven by low-grade chronic inflammation, an altered hepatic acute-phase response, gut dysbiosis and bacterial translocation, neutrophil dysfunction due to insulin resistance, reduced lipopolysaccharide (LPS) inactivation, and impaired pathogen clearance [14]. Notably, MASLD might dysregulate the “liver–lung axis”—the bidirectional network in which alveolar macrophages secrete cytokines (e.g., IL-1, IL-6, and TNF-α) during pneumonia triggering the liver acute-phase response, which in turn modulates lung immune responses [15]. However, the immune response to infection in MASLD patients remains unexplored, and specific mediators driving liver–lung crosstalk are not yet fully characterized.

Semaphorins (SEMA), a family of glycoproteins, have recently emerged as key modulators of immune and inflammatory responses [16,17]. While most evidence comes from animal studies, the diagnostic and therapeutic potential of SEMA is increasingly recognized in immune-mediated diseases and cancer [18,19,20]. Only a few studies have examined their role in infections. Serum SEMA3C, SEMA5A and SEMA6D levels correlate with liver fibrosis stage in chronic hepatitis C [21]. In COVID-19, SEMA3F and SEMA7A were associated with COVID-19 complications [22]. In our sepsis cohort, SEMA kinetics were linked to sepsis complications with SEMA3A, SEMA3C, SEMA4D and SEMA7A associated with mortality [23]. Furthermore, we recently demonstrated that SEMA are potential new biomarkers of MASLD, with serum concentrations correlating with both steatosis and the fibrosis stage [24].

Despite their biological relevance, the interaction between semaphorin signaling, cytokine dynamics, and MASLD has not yet been investigated in the context of sCAP. To address this gap, we conducted a prospective immune profiling study in adults hospitalized with sCAP, comparing patients with and without MASLD, and examined its relationship to liver disease parameters and clinical outcomes. We hypothesized that MASLD would be associated with a distinct semaphorin–cytokine immune signature. This study provides, to our knowledge, the first integrated characterization of semaphorin and cytokine dynamics in patients with MASLD during sCAP, offering novel insights into host–metabolic–immune interactions with potential implications for biomarker development and future immunomodulatory strategies.

## 2. Results

Between December 2023 and November 2024, 404 adults hospitalized with CAP were screened. After applying eligibility criteria and confirming biospecimen availability, 108 patients were enrolled. Among these, 54 (50.0%) met predefined diagnostic criteria for MASLD, based on hepatic steatosis confirmed by vibration-controlled transient elastography (VCTE) and at least one predefined cardiometabolic risk factor (Figure 1).

### 2.1. Study Cohort and MASLD-Associated Characteristics

Baseline demographic and clinical characteristics are presented in Table 1. The median age was 61 years (IQR 44–74), with a predominance of male patients (70%). Compared to non-MASLD counterparts, MASLD patients had significantly higher body mass index (BMI, 32.1 vs. 25.6 kg/m^2^, *p* < 0.0001), waist circumference (WC, 105 vs. 93 cm, *p* < 0.0001), and waist-to-height ratio (WHtR, 0.63 vs. 0.57, *p* < 0.0001). Central obesity (defined as WHtR ≥ 0.5) was present in 85.1% of MASLD patients versus 62.9% of non-MASLD patients (*p* = 0.0148).

The burden of comorbidities, including type 2 diabetes mellitus (T2DM), cardiovascular disease, chronic kidney disease, and chronic pulmonary conditions, did not significantly differ between groups. Similarly, pneumonia severity indices at admission—including SOFA, PSI, SMART-COP, and CURB-65 scores—were comparable between MASLD and non-MASLD patients.

VCTE performed within 72 h of admission revealed more advanced hepatic involvement in the MASLD group. Controlled attenuation parameter (VCTE-CAP) values were significantly elevated (297 vs. 203 dB/m, *p* < 0.0001), as were liver stiffness measurement (LSM) values (6.3 vs. 5.5 kPa, *p* = 0.032). The FAST score, an integrative marker of steatosis, stiffness, and AST levels, was also markedly higher in MASLD patients (median 0.52 vs. 0.26, *p* = 0.0002), with 63% exceeding the clinically relevant threshold of 0.35 [25] (Figure 1, panel B).

Routine laboratory findings at admission were largely comparable between groups, except for peripheral blood leukocyte (WBC) count, which was significantly higher in MASLD patients (13 vs. 8.3 × 10^9^/L, *p* = 0.012). Inflammatory markers (CRP, PCT), liver enzymes, renal function, and coagulation parameters did not differ significantly (Appendix A).

Chest imaging revealed multilobar involvement in 68% and bilateral infiltrates in 55% of patients, with no significant differences between groups. Pleural effusions were present in 47% of cases (Appendix A). The distribution of pathogens—including *Streptococcus pneumoniae* (*n* = 4), *Legionella pneumophila* (*n* = 13), *Mycoplasma pneumoniae* (*n* = 11), and respiratory viruses such as influenza (*n* = 25) and adenovirus (AdV) (*n* = 16)—did not differ significantly between the groups (Appendix A).

### 2.2. Clinical Presentation, Early Course, and Outcomes

At admission, MASLD patients presented with more severe respiratory symptoms; tachypnoea was present in all MASLD cases (100% vs. 82%, *p* = 0.001), with higher heart rates (104 vs. 93 beats/min, *p* = 0.008), and elevated PaCO_2_ (42 vs. 35 mmHg, *p* = 0.013). Despite these physiologic differences, no significant variation was observed in standard pneumonia severity scores between groups and SpO_2_/FiO_2_ ratios (242 vs 241, *p* = 0.665), as shown in Appendix A.

During hospitalization, MASLD patients had significantly higher rates of invasive mechanical ventilation (IMV) (OR 3.1, 95% CI 1.1–8.7) and vasopressor-requiring shock (OR 4.0, 95% CI 1.3–12). Continuous renal replacement therapy (CRRT) was used more frequently in MASLD patients (OR 2.5, 95% 0.63–9.3), though this did not reach statistical significance. Time to clinical stability was slightly longer in MASLD patients (median 7 vs. 6 days, *p* = 0.63), while length of stay and in-hospital mortality were similar between groups (Table 2 and Appendix A).

To further assess the prognostic value of hepatic phenotype, the FAST score was analyzed in relation to clinical outcomes. Higher admission FAST scores were associated with increased risk of IMV, CRRT, and in-hospital mortality, as demonstrated by ROC curve analysis (Appendix A). In addition, FAST score correlated with time to clinical stability (Appendix A), and higher scores were linked to delayed stabilization, as shown by Kaplan–Meier survival analysis. In multivariable logistic regression models, MASLD remained independently associated with the need for IMV (OR 3.8, 95% CI 1.2–12.4, *p* = 0.027), as well as FAST score > 0.5 (OR 7.8, 95% CI 2.1–29.5, *p* = 0.002) (Appendix A).

### 2.3. Circulating Immune Profile at Hospital Admission

We measured serum concentrations of 13 cytokines upon hospital admission to analyze the initial immune response in MASLD compared to non-MASLD patients. At admission, MASLD patients exhibited a distinct circulating cytokine profile compared to non-MASLD patients. Specifically, MASLD patients showed significantly higher serum concentrations of IL-2 (median 6.3 vs. 3.7 pg/mL, *p* = 0.043), IL-17A (12.4 vs. 6.0 pg/mL, *p* = 0.035), CXCL10 (3130 vs. 1151 pg/mL, *p* = 0.015), and TGF-β1 (196 vs. 100 pg/mL, *p* = 0.024), as well as the anti-inflammatory cytokine IL-10 (18.7 vs. 10.0 pg/mL, *p* = 0.046). No significant differences were observed for other measured cytokines, including IL-6, IL-1β, IL-4, TNF-α, IFN-γ, IL-8, IL-12p70 and CCL2. These findings indicate that MASLD is associated with distinct admission levels of chemotactic, pro-inflammatory, and regulatory cytokines (Appendix A, Figure 2).

### 2.4. Temporal Dynamics of Cytokines During Early Disease Course

To investigate the effect of MASLD on the early immune response, we compared cytokine levels at hospital admission (day 1) and on day 5.

Patients with MASLD exhibited more pronounced temporal changes in circulating immune mediators (Appendix A). IL-6 concentrations declined significantly in MASLD patients by 821 pg/mL (95% CI 128–1514), compared to a 729 pg/mL reduction in non-MASLD patients (95% CI 35–1422), reflecting early attenuation of systemic inflammation.

IL-10 decreased by 15 pg/mL in MASLD patients (95% CI 6.8–23) and by 10 pg/mL in non-MASLD patients (95% CI 2.3–18). This reduction indicates a more pronounced resolution of early regulatory signaling within the MASLD group.

Significant reductions were also observed in T-cell–associated cytokines in MASLD patients. IL-17A declined by 5.8 pg/mL (95% CI 2.0–9.5), while no significant change was observed in non-MASLD patients (Δ = 0.3 pg/mL, 95% CI −3.3 to 3.9). Similarly, IFN-γ decreased by 24 pg/mL in MASLD patients (95% CI 2.6–45), with a non-significant reduction in non-MASLD patients (Δ = 19 pg/mL, 95% CI −1.8 to 41).

MASLD was also associated with more pronounced changes in chemotactic signaling. CXCL10 levels decreased by 2805 pg/mL in MASLD patients (95% CI 1597–4013) versus 1903 pg/mL in non-MASLD patients (95% CI 695–3111). In contrast, MCP-1 (CCL2) declined significantly only in non-MASLD patients (Δ = 723 pg/mL, 95% CI 79–1367), while the change in MASLD patients was not significant (Δ = 506 pg/mL, 95% CI −139 to 1150).

TGF-β1, associated with immune regulation and tissue remodeling, increased over time in both groups. The rise was significant in non-MASLD patients (Δ = 91 pg/mL, 95% CI 21–161), while MASLD patients showed a non-significant trend toward increase (Δ = 69 pg/mL, 95% CI −149 to 11).

Collectively, these data demonstrate that MASLD is associated with an amplified and rapidly evolving early immune response, and this kinetic profile may reflect altered inflammatory resolution or immunometabolic regulation in the context of MASLD (Figure 2).

### 2.5. Semaphorin Kinetics and Lung–Systemic Compartmentalization

To characterize early SEMA responses, we measured serum concentrations of SEMA3A, 3F, 4D, 5A, and 7A at hospital admission (day 1) and on day 5 in patients with and without MASLD (Figure 3 and Appendix A).

At admission, MASLD patients had significantly higher serum levels of SEMA7A (median 31 vs. 28 ng/mL; Δ = +4.6 ng/mL, 95% CI 0.14–9.0, *p* = 0.043) and SEMA3A (16 vs. 15 ng/mL; Δ = +1.55 ng/mL, 95% CI 0.06–3.04, *p* = 0.041), and significantly lower levels of SEMA5A (15 vs. 24 ng/mL; Δ = −15 ng/mL, 95% CI −26 to −3.2, *p* = 0.012) compared to non-MASLD patients. No significant between-group differences were observed for SEMA3F or SEMA4D at baseline.

Kinetic analysis revealed that SEMA7A levels decreased significantly from day 1 to day 5 in both groups: by 6.7 ng/mL in MASLD patients (95% CI 3.7–9.7, *p* < 0.0001) and by 3.3 ng/mL in non-MASLD patients (95% CI 0.33–6.3, *p* = 0.030). SEMA3A decreased significantly in MASLD patients (Δ = −1.63 ng/mL, 95% CI 0.53–2.72, *p* = 0.004), while it increased in non-MASLD patients (Δ = +1.14 ng/mL, 95% CI 2.2 to 0.09, *p* = 0.004).

SEMA5A levels remained stable in MASLD patients (Δ = −0.08 ng/mL, 95% CI −5.9 to 6.0, *p* = 0.978) and showed a non-significant trend toward decrease in non-MASLD patients (Δ = −5.7 ng/mL, 95% CI −0.45 to 12, *p* = 0.069). SEMA4D increased over time in non-MASLD patients (Δ = −6.6 ng/mL, 95% CI −13 to −0.60, *p* = 0.031), while no significant change was observed in MASLD (Δ = −2.9 ng/mL, 95% CI −9.1 to 3.2, *p* = 0.349). SEMA3F concentrations remained unchanged in both groups.

To assess compartment-specific immune signaling, in intubated patients we measured concentrations of selected semaphorins in paired serum and bronchoalveolar lavage fluid (BALF) samples collected at the time of intubation (Appendix A). All measured semaphorins were detectable in BALF, with distinct compartmental distributions. SEMA3F concentrations were significantly higher in serum compared to BALF (median 9.4 vs. 1.2 ng/mL; *p* = 0.009), as were SEMA4D (53.0 vs. 3.2 ng/mL; *p* = 0.006) and SEMA5A (17.0 vs. 0.7 ng/mL; *p* = 0.007). SEMA3A concentrations were low in BALF and not quantitatively comparable. In contrast, SEMA7A levels were significantly elevated in BALF compared to serum (532 vs. 29 ng/mL; *p* = 0.003), indicating compartmental enrichment.

Furthermore, to explore functional relationships, we performed correlation analysis between SEMA and cytokines in BALF (Appendix A). SEMA4D correlated strongly with CXCL10 (r = 0.90, *p* = 0.02) and IL-6 (r = 0.82, *p* = 0.04); SEMA7A correlated with CXCL10 (r = 0.68, *p* = 0.02) and CCL2 (r = 0.73, *p* = 0.01); and SEMA3F correlated with IL-6 (r = 0.83, *p* = 0.04). IL-6 in BALF negatively correlated with serum SEMA3A (r = −0.98, *p* < 0.01), while CXCL10 in serum correlated with BALF CCL2 (r = 0.66, *p* = 0.03). Additional associations were observed with clinical parameters. WHtR was positively associated with BALF concentrations of SEMA5A (r = 0.93, *p* < 0.01) and SEMA7A (r = 0.70, *p* = 0.04). BALF IL-6 correlated positively with liver steatosis grade (r = 0.73, *p* = 0.03), while SEMA4D in BALF showed a strong inverse correlation with steatosis (r = −1.00, *p* < 0.01). WBC counts negatively correlated with BALF SEMA3F (r = −0.96, *p* = 0.01) and SEMA5A (r = −0.60, *p* = 0.04).

### 2.6. Semaphorin–Cytokine Interactions and Relationship to Liver Disease Severity

To better understand the immunometabolic context of SEMA signaling in pneumonia, we analyzed correlations between serum SEMA levels, disease severity, liver involvement, cytokine profiles, and peripheral immune cell subsets (Figure 4).

At hospital admission, serum SEMA7A levels showed strong positive correlations with clinical severity scores, including SOFA (r = 0.43), PSI (r = 0.34), and SMART-COP (r = 0.51). SEMA3F (r = 0.25) and SEMA5A (r = 0.16) also correlated with SOFA and SMART-COP (r = 0.27), although to a lesser extent.

Several semaphorins were associated with liver disease severity: SEMA3A (r = 0.32) and SEMA7A (r = 0.18) with VCTE-CAP; SEMA4D (r = 0.20) and SEMA7A (r = 0.18) with LSM; and SEMA3F (r = 0.22) and SEMA7A (r = 0.26) with FAST score. In addition, SEMA4D (r = 0.31) and SEMA7A (r = 0.30) correlated positively with AST and ALT levels.

SEMA3F showed positive correlations with CRP (r = 0.22), WBC count (r = 0.33), and neutrophil-to-lymphocyte ratio (NLR) (r = 0.23), while SEMA5A was negatively associated with CRP (r = −0.18). Correlations with cytokines revealed negative associations between SEMA3F and IL-2 (r = −0.26), IL-10 (r = −0.19), and IL-12p70 (r = −0.19); SEMA4D and SEMA5A both correlated negatively with IL-17A (r = −0.20 and −0.19, respectively), whereas SEMA4D correlated positively with TNF-α (r = 0.24), and SEMA7A with IL-8 (r = 0.29).

By Day 5, the correlation network evolved. SEMA7A remained positively associated with SOFA (r = 0.37) and FiO_2_ (r = 0.37), and inversely with SpO_2_/FiO_2_ (r = −0.21), while SEMA3A showed the opposite pattern (SOFA r = −0.27; FiO_2_ r = 0.32; SpO_2_/FiO_2_ r = 0.25). SEMA7A also remained associated with LSM (r = 0.16) and FAST score (r = 0.21). SEMA3F correlated strongly with IL-2 (r = 0.86) and TNF-α (r = 0.73), and more moderately with IL-8 (r = 0.47) and IL-10 (r = 0.36). SEMA5A correlated with IFN-γ (r = 0.30), SEMA4D with IL-8 (r = 0.26), and SEMA7A with TGF-β1 (r = 0.19).

To further characterize the immunological context, SEMA correlations with peripheral blood lymphocyte subsets were analyzed (Appendix A). SEMA7A positively correlated with the relative proportion of B cells (r = 0.23) and negatively with NK cells (r = −0.19). SEMA3A was associated with CD4^+^ T-cell count (r = 0.27) and NK-cell proportion (r = 0.23). SEMA3F correlated positively with CD4^+^ and CD8^+^ T cell proportions (r = 0.23 for both), and negatively with activated CD8^+^ T cells (r = −0.24). SEMA4D was positively associated with activated T cells (r = 0.23).

### 2.7. Pathogen-Specific Modulation of Immune Responses

Given that both microbial etiology and underlying MASLD may influence the initial immune response, we compared baseline cytokine concentrations between MASLD and non-MASLD patients across five etiological subgroups (Figure 5).

In typical bacterial pneumonia, IL-6 was elevated in MASLD patients at admission (Δ = +2289 pg/mL; 95% CI 95–2283), as was TGF-β1 (Δ = +186 pg/mL; 95% CI 2.7–369). In the *Legionella pneumophila* subgroup, MASLD was associated with a marked increase in IL-17A (median 19.8 vs 2.4 pg/mL; Δ = +24.8; 95% CI 4.2–45.5). In *Mycoplasma pneumoniae*, MASLD patients showed higher levels of IL-17A (Δ = +16.2; 95% CI 5.4–26.9) and IL-10 (Δ = +12.9; 95% CI 2.1–23.8). Among influenza virus infections, MASLD was associated with significantly higher CXCL10 (Δ = +3066; 95% CI 1156–4976), IL-10 (Δ = +24.6; 95% CI 2.5–46.7), IL-17A (Δ = +15.0; 95% CI 4.3–25.7), and TGF-β1 (Δ = +148; 95% CI 9.8–286).

To determine whether MASLD modifies pathogen-specific immune trajectories, we calculated the day-1 minus day-5 change (Δ) for major cytokines (CXCL10, IL-17A, TGF-β1) and three semaphorins (SEMA3A, SEMA4D, SEMA7A) within each etiological group, as shown in Figure 6, and Appendix A.

CXCL10 significantly decreased in the influenza MASLD group (Δ = 3333, 95% CI 1368-5299), and the difference compared to non-MASLD was found to be significant (*p* = 0.0167). Similarly, IL-17A significantly decreased in bacterial pneumonia (Δ = 8.6, 95% CI 1.4–15.0). IL-17A showed the largest decline differences between MASLD and non-MASLD patients in *Legionella* (*p* = 0.0081), influenza (*p* = 0.0402) and *M. pneumoniae* (*p* = 0.0129). IL-10 significantly decreased in MASLD patients with AdV (Δ = 29.3, 95% CI 7.7–52). SEMA3A significantly decreased in MASLD bacterial pneumonia (Δ = 1.8, 95% CI 0.5–3.1), and non-MASLD influenza group (Δ = 7.3, 95% CI 5.9–8.7). TGF-β1 significantly increased in non-MASLD patients with bacterial pneumonia (Δ = −168, 95% CI 5–331) and MASLD group with *M. pneumoniae* (Δ = −564, 95% CI −793–−335) and AdV (Δ = −300, 95% CI −552–−48).

In MASLD, SEMA4D significantly increased in influenza (Δ = 11.5, 95% CI −4.2–27.2), but increased in bacterial pneumonia (Δ = −2.8, 95% CI −11.4–5.7). Significant changes were observed in the MASLD group in SEMA7A decline in *Legionella* (Δ = 17.3, 95% CI 8–26), bacterial pneumonia (Δ = 6.4, 95% CI 0.7–12.2) and influenza (Δ = 9.8, 95% CI 5.1–14.6).

Two-way ANOVA confirmed significant Time × Etiology interactions for CXCL10, IL-17A, TGF-β1, SEMA3A, SEMA4D, and SEMA7A (all FDR-adjusted *p* < 0.05), with MASLD acting as an additional modifier for CXCL10, IL-17A, SEMA3A, and SEMA7A. These kinetics are visualized in Figure 6, and full per-pathogen panels are provided in Appendix A.

### 2.8. Associations Between Immune Markers and Clinical Outcomes

To explore the prognostic relevance of immune markers in sCAP, we examined associations between cytokines, semaphorins, and clinical outcomes including IMV, vasopressor-requiring shock, CRRT, and in-hospital mortality.

Patients requiring IMV had significantly higher admission concentrations of CXCL10 (Δ = +3272, 95% CI 1664–4880), IL-10 (Δ = +29.4, 95% CI 18.3–40.4), IL-8 (Δ = +17.6, 95% CI 0.6–34.5), CCL2 (Δ = +901, 95% CI 291–1512), and TGF-β1 (Δ = +96.2, 95% CI 30.3–162.1), and showed marked declines in CXCL10, IL-10, CCL2 and IL-1β by Day 5 (Appendix A). In contrast, patients managed with conventional oxygen therapy exhibited significant Day 5 reductions in CXCL10, IL-10, IL-6, IL-17A, and IFN-γ, while maintaining lower levels of TGF-β1 and IL-8 throughout (Appendix A).

SEMA dynamics also differed by IMV status. SEMA7A concentrations were significantly elevated at baseline in the IMV group (Δ = +8.9, 95% CI 3.5–14.4) and declined significantly over time in both groups (Figure 7). However, IMV status accounted for substantial variance in two-way ANOVA. SEMA5A levels did not differ at baseline but were significantly higher by Day 5 in the IMV group (Δ = +18.8 ng/mL, 95% CI 4.6–33.0). Longitudinally, SEMA5A increased in IMV patients (Δ = +8.2, *p* = 0.081) and decreased in non-IMV patients (Δ = −5.4, *p* = 0.019), indicating divergent kinetic trajectories. In ROC analysis, admission and day 5 SEMA7A levels, as well as day 5 SEMA5A (AUC = 0.75, 95% CI 0.66–0.84), showed good predictive performance for IMV requirement (Figure 7).

Similarly, patients who developed acute kidney injury (AKI) requiring CRRT had elevated baseline concentrations of CXCL10 (Δ = 3236, 95% CI 1014–5459), IL-10 (Δ = 48.6, 95% CI 34–63), IL-8 (Δ = 39.9, 95% CI 17.7–62.1) and IL-6 (Δ = 1441, 95% CI 125–2697). While CXCL10 and IL-10 significantly decreased in both groups, in the non-CRRT group the decline was significantly lower (Figure 8, Appendix A). Both IL-10 and CXCL10 showed good sensitivity and specificity for detecting need for CRRT in ROC analysis (Figure 8). While TGF-β1 concentrations did not significantly differ between CRRT and non-CRRT groups at either timepoint, there was a significant increase from Day 1 to Day 5 observed exclusively in the non-CRRT group (Δ = +87.1 pg/mL, *p* = 0.0023), while levels remained unchanged in CRRT patients, thus suggesting impaired or delayed resolution of TGF-β1 signaling in the context of AKI. Similarly, IL-6 (Δ = −701 pg/mL, *p* = 0.0078) and CCL2 (MCP-1) (Δ = −441 pg/mL, *p* = 0.0123) levels significantly decreased only in the non-CRRT group, while remaining unchanged in CRRT patients (Appendix A). Furthermore, patients requiring CRRT had elevated baseline concentrations of SEMA3F (Δ = +2.75, 95% CI 0.26–5.24) and SEMA7A (Δ = +15.8, 95% CI 8.4–23.2), both of which declined significantly by day 5. SEMA4D kinetics also differed between groups; while it decreased in CRRT patients (Δ = −15.7, *p* = 0.0387), in non-CRRT SEMA4D increased by day 5 (Δ = +6.2, *p* = 0.004). Subsequent ROC analysis identified SEMA3F and SEMA7A thresholds predictive of CRRT initiation (Figure 7).

Patients who developed shock had significantly higher baseline SEMA7A concentrations (Δ = 8.3, 95% CI 2.3–14.4), accompanied by a marked decrease in SEMA3A (Δ = −2.7, *p* = 0.01), SEMA3F (Δ = −1.9, *p* = 0.03), and SEMA7A (Δ = −6.2, *p* = 0.02) levels by day 5. SEMA4D kinetics also differed; only patients without shock had increase in SEMA4D levels by day 5 (Δ = +6.4, *p* = 0.006). ROC analysis identified SEMA3A and SEMA3F thresholds associated with subsequent shock (Figure 7). Cytokine profiling showed that patients with shock had higher baseline CXCL10 (Δ = +1965, 95% CI 168–3761), IL-10 (Δ = +23.6, 95% CI 11.3–36.1), and CCL2 (Δ = +889, 95% CI 223–1555) with a decline in CCL2 over time (Δ = −1018, *p* = 0.01). In contrast, patients without shock showed decreasing CXCL10 (Δ = −2264, *p* < 0.01), IL-10 (Δ = −9.3, *p* = 0.001), IL-6 (Δ = −715, *p* = 0.009), and IL-8 (Δ = −10.6, *p* = 0.04), and a significant increase in TGF-β1 (Δ = +81.9, *p* = 0.005) by Day 5 (as shown in Figure 8, and Appendix A).

We further explored the prognostic relevance of SEMA kinetics by stratifying patients according to survival status (Appendix A). On admission, SEMA4D levels were significantly higher in non-survivors compared to survivors (Δ = +26.7 ng/mL; 95% CI 6.1–47.2; *p* = 0.012). Two-way repeated measures ANOVA identified significant time × mortality interactions for both SEMA4D (*p* = 0.0004) and SEMA7A (*p* = 0.032), indicating divergent temporal trajectories between outcome groups. SEMA4D levels decreased significantly in non-survivors (Δ = −22.5 ng/mL; 95% CI 7.3–37.7; *p* = 0.0042), while survivors exhibited a modest but significant increase (Δ = +6.8 ng/mL; 95% CI 2.7–10.9; *p* = 0.0015). Similarly, SEMA7A declined in both groups, but the magnitude of reduction was greater in non-survivors (Δ = −13.6 ng/mL; 95% CI 5.4–21.8; *p* = 0.0014) compared to survivors (Δ = −4.3 ng/mL; 95% CI 2.1–6.4; *p* = 0.0002). In survival analysis, admission SEMA4D > 50 ng/mL (HR 9.6, 95% CI 1.0–93.3, *p* < 0.001) and ΔSEMA7A > 6.5 ng/mL (HR 6.2, 95% CI 1.2–31.7, *p* = 0.0122) were associated with significantly increased risk of death. These findings suggest that dysregulated resolution of semaphorin signaling, particularly involving SEMA4D and SEMA7A, may be associated with adverse outcomes and impaired immunological recovery in sCAP.

## 3. Discussion

In this prospective cohort of adults hospitalized with sCAP, we demonstrate that MASLD is not just a passive comorbidity but a central immunometabolic modulator of disease trajectory. MASLD independently predicted early respiratory and circulatory failure, even in the context of similar baseline severity scores. Most notably, our study is the first to show that MASLD shapes pathogen-specific immune responses through a distinctive cytokine–semaphorin signature. These findings position MASLD as a dynamic amplifier of infection and support its inclusion in risk stratification frameworks and possibly therapeutic targeting strategies.

The prevalence of MASLD of 50% in our cohort is significantly higher than estimated in the general population (30%) [7], suggesting it may predispose patients to the development of sCAP. Our data demonstrates that MASLD is independently associated with development of acute respiratory failure and shock, even after adjusting for age, T2DM, and baseline illness severity. Earlier observational studies which linked MASLD to risk or adverse outcomes in CAP lacked biological correlates [11,12,13]. Our findings addressed this gap, revealing that MASLD confers a unique immunological phenotype with functional consequences for host defense and subsequent organ dysfunction. Our results provide support for the concept of the liver–lung axis during pneumonia. A significantly higher FAST score (0.52 vs 0.26), which reflects the combination of steatosis, fibrosis, and AST values, as well as the presence of MASLD per se, was associated with worse clinical outcomes, including the need for IMV, CRRT, and prolonged time to clinical stabilization.

At admission, MASLD patients displayed an altered systemic cytokine profile, with significantly elevated IL-17A, IL-2, CXCL10, TGF-β1 and IL-10. Simultaneous hyperactivation of both pro-inflammatory (IL-17A, CXCL10) and immunoregulatory (IL-10, TGF-β1) pathways may represent impaired ability to resolve inflammation. Kinetics of CXCL10 and IL10 were also associated with sCAP complications. These might mirror the chronic low-level inflammation associated with MASLD which may prime the immune system for an exaggerated response to infection. Prior human studies in MASLD have shown increased baseline IL-6, IL-17A and TGF-β1 which are associated with hepatic inflammation, insulin resistance, and fibrosis [26,27]. IL-17A drives neutrophilic inflammation and epithelial injury and plays a crucial role in host defense against bacterial and fungal infections, particularly at mucosal surfaces, but its increased expression can exacerbate lung damage [28]. CXCL10, a key interferon-induced chemokine which attracts activated T-cells, NK cells, and monocytes to the site of infection, is implicated in acute lung injury and correlates with poor pneumonia outcomes [29]. The simultaneous increase in TGF-β1 and IL-10, while considered classically immunosuppressive, if triggered prematurely can suppress antimicrobial responses [30]. Paradoxically, while IL-10 plays a protective role by limiting tissue damage, elevated levels can reflect a compensatory anti-inflammatory response that impairs host defense, as commonly observed in early sepsis as a marker of severity and impending immunosuppression [31]. Collectively, these findings suggest that MASLD shapes the immune landscape toward a Th1/Th17-prone hyperinflammatory state, accompanied by elevated anti-inflammatory mediators.

One of the central findings of our study is the temporally dysregulated, biphasic cytokine trajectory marked by early cytokine surges followed by premature immunological attenuation observed in MASLD. MASLD patients demonstrated markedly steeper decline in serum levels of IL-17A, CXCL10, IL-6, IFN-γ and IL-10 when compared to non-MASLD patients. Notably, levels of TGF-β1—an essential mediator of immune regulation and tissue remodeling—increased significantly only in the non-MASLD group, indicating impaired activation of late-phase immunoregulatory programs. The observed pattern mirrors the biphasic immunopathology seen in sepsis, where MASLD patients exhibited higher baseline levels of IL17A, IL-23, IL-33, CXCL10 and TGF-β1, yet experienced steeper declines in in IL-10, IL-23, CXCL10 and TGF-β1 in comparison to non-MASLD counterparts [32]. This concept is also supported in animal models of lung infections where metabolic state predisposes to an exaggerated early cytokine response, more severe neutrophil-driven lung damage, impaired bacterial clearance and worse outcomes, suggesting an attenuated resolution phase and failure of tissue repair [33,34]. In addition, several pathophysiological mechanisms in MASLD likely contribute to this maladaptive trajectory, including disrupted liver acute-phase responses, mitochondrial dysfunction, neutrophil dysfunction and sustained NF-κB activation in hepatic and myeloid cells [35].

Furthermore, MASLD was associated with distinct pathogen-specific immune responses. While IL-6 and TGF-β1 were significantly elevated in MASLD patients with typical bacterial sCAP at admission, in MASLD patients with *L. pneumophila*, *M. pneumoniae* and influenza sCAP we observed a sharp early increase and a significant decline in IL-17A five days later. Although elevated IL-17A levels have been linked to poor pneumonia outcomes, this cytokine is important for the clearance of intracellular pathogens and its premature collapse may hinder host defense [36,37,38]. In influenza, MASLD patients also displayed exaggerated interferon-induced chemokine CXCL10 and immunoregulatory TGF-β1 and IL-10 responses, possibly enhancing tissue damage and impairing viral clearance. These findings support a model where MASLD acts as a pathogen-sensitive amplifier of immune dysfunction.

To the best of our knowledge, there are only two published human studies on SEMA profiling in pneumonia patients. One examined COVID-19 patients, and the other ARDS patients [22,39]. Given their established roles in immune cell interactions and their potential as biomarkers in sepsis and respiratory infections, we hypothesized that semaphorins would provide novel insights into the immunopathogenesis of sCAP, particularly in the context of MASLD where they correlated with liver phenotype [24].

Patients with MASLD had significantly higher serum SEMA7A and SEMA3A, and significantly lower SEMA5A concentrations on admission. While both groups had a significant decline of serum SEMA7A concentrations at the 5th day of hospitalization, SEMA3A concentrations significantly declined in MASLD patients and increased in non-MASLD patients. SEMA4D showed a significant increase only in non-MASLD patients.

SEMA7A is important throughout the initial stages of inflammation and has an important pulmonary immunomodulatory effect [39]. In murine lung injury models, SEMA7A is induced by pro-inflammatory cytokines and in turn amplifies pulmonary inflammation and neutrophil migration [40]. Importantly, we showed the SEMA7A compartmentalization in BALF, highlighting its role regulating pulmonary inflammation. This parallels observations in rheumatoid arthritis, where SEMA7A levels in serum and synovial fluid correlated with disease activity [41]. SEMA7A surges were also reported in patients with sepsis [23], COVID-19 [22] and ARDS [39]. The consistent elevation of SEMA7A across different inflammatory conditions suggests that SEMA7A may represent a common pathway in pathological immune activation. In addition, correlation of SEMA7A BALF concentrations with central obesity point to the possible SEMA-mediated metabolic effect on lung inflammation. SEMA7A is highly expressed in activated hepatic stellate cells (which is even more pronounced in liver injury) and drives fibrogenesis and inflammation, including upregulating CCL2 to recruit monocytes [42]. Therefore, it is not surprising that in our study SEMA7A showed positive correlations with liver disease severity—VCTE-CAP, LSM, FAST score, AST and ALT levels.

SEMA3A is known to increase endothelial permeability and neutrophil transmigration and SEMA5A strongly enhances endothelial cell proliferation and migration while inhibiting apoptosis [43]. The initial SEMA3A surge and SEMA5A deficiency in MASLD patients could weaken vascular repair and resolution [44]. Together, these SEMA alterations could link liver inflammation to lung injury. Conversely, SEMA4D is known to support endothelial repair and anti-inflammatory polarization—SEMA4D evokes angiogenic responses and repels monocyte migration [45]. The lack of SEMA4D upregulation in MASLD (seen in our data, compared to rising SEMA4D in non-MASLD) suggests a failure to engage in a reparative process.

Finally, semaphorins emerged as robust predictors of sCAP severity. SEMA7A and SEMA4D levels and kinetics were associated with need for IMV, shock, AKI requiring CRRT and mortality. In addition, serum SEMA7A concentrations measured at admission correlated strongly with SOFA, PSI and SMART-COP scores. In patients who developed organ dysfunction, SEMA7A failed to decrease by day 5, in contrast to patients without it, implying an inadequate down-regulation of inflammation. BALF–serum concordance of CXCL10, CCL2, and IL-8 with SEMA7A further highlights its relevance to lung injury. Conversely, SEMA4D failed to increase in patients with shock and IMV, and even decreased in patients requiring CRRT, once again highlighting the possibly important role of SEMA4D in controlling dysregulated inflammation.

In summary, our data indicate that MASLD shapes a distinct immunokinetic pattern in sCAP, characterized by an exaggerated early inflammatory burst followed by insufficient reparative signaling. This trajectory likely reflects the interplay between metabolic dysfunction—marked by insulin resistance, hepatic lipotoxicity, and chronic low-grade inflammation—and dysregulated immune pathways. Semaphorins such as SEMA7A and SEMA4D, which regulate leukocyte trafficking, macrophage activation, and fibroproliferative signaling, correlated in our study with both pro-inflammatory (CXCL10) and regulatory (TGF-β1) mediators, suggesting a role in modulating the transition from inflammation to repair. In MASLD, altered semaphorin–cytokine crosstalk could prematurely dampen pro-resolving signals, impair tissue remodeling, and act within a disrupted liver–lung axis, where hepatic dysfunction limits effective pulmonary immune regulation. These mechanisms may underlie the insufficient immune repair we observed and highlight potential targets for immunomodulation in metabolically vulnerable patients.

This study has several limitations. First, it was conducted at a single center with a modest sample size, which may limit generalizability. Although patients were carefully phenotyped and longitudinally followed, the observational design precludes causal inference, and residual confounding by unmeasured variables cannot be excluded. Immune profiling was limited to two timepoints (day 1 and day 5), which may not fully capture longer-term immunological trajectories. Cytokine and semaphorin analyses were exploratory and not adjusted for BMI or other metabolic syndrome components, both to avoid over-adjustment and due to limited statistical power in subgroup comparisons. In-hospital mortality in our cohort was lower than the 20–30% typically reported for ICU-treated sCAP. This difference is likely multifactorial and may reflect the inclusion of both ICU and non-ICU patients, the predominance of patients without advanced comorbidity burden, and early targeted antimicrobial and supportive care in our center. Nevertheless, mortality findings should be interpreted with caution given potential differences in case mix and healthcare setting compared to prior ICU-focused cohorts. In addition, BALF analyses were restricted to intubated patients, which may reflect immune responses in those with more severe respiratory failure and may not be representative of the entire sCAP cohort. Future studies should include BALF or other lower respiratory tract sampling from both intubated and non-intubated patients to better characterize differences between local and systemic immune responses. The pathogen-stratified analyses were exploratory, and some subgroups had small sample sizes, which may limit statistical power; similarly, the observed associations between semaphorins, cytokines, and outcomes should be interpreted with caution and considered hypothesis-generating until confirmed in larger, multi-center studies with mechanistic follow-up.

Nevertheless, our findings underscore the relevance of MASLD as an important determinant of pneumonia severity and immune response phenotype. By identifying a distinct MASLD-associated immunological signature, our study supports the development of precision medicine tools for early risk stratification. In conclusion, this study identifies MASLD as an immunologically active determinant of pneumonia severity and introduces semaphorins as novel mediators of liver–lung immune crosstalk.

## 4. Materials and Methods

### 4.1. Study Design and Population

A prospective, non-interventional, single-center observational study was carried out at the University Hospital for Infectious Diseases (UHID) in Zagreb, Croatia (SepsisFAT, ClinicalTrials.gov identifier NCT06021743). All consecutively hospitalized adults (≥18 years) with CAP (radiologically confirmed, symptomatic pneumonia not acquired in hospital and diagnosed within 48 h of admission) between December 2023 and November 2024 were screened. Patients were enrolled if, within the first 48 h of hospitalization, they met ≥1 major (septic shock requiring vasopressors; respiratory failure requiring HFNO with FiO_2_ ≥ 50% and PaO_2_/FiO_2_ ≤ 300 mmHg) or non-invasive ventilation [NIV] or IMV; arterial pH < 7.30 and/or ≥3 minor modified sCAP criteria (respiratory rate ≥ 30/min; PaO_2_/FiO_2_ ≤ 250 mmHg; multilobar infiltrates; confusion/disorientation; elevated blood urea nitrogen (>8.3 mmol/L); WBC ≤ 4 × 10^9^/L; platelets ≤ 100 × 10^9^/L or >400 × 10^9^/L; hypothermia (T ≤ 36 °C); hypotension requiring aggressive fluid resuscitation (≥20 mL/kg in 2 h or ≥30 mL/kg in 3 h)) [46,47]. All participants gave written informed consent. The study conformed to the ethical guidelines of the Declaration of Helsinki and was approved by the Ethics Committee of the UHID Zagreb.

Predefined exclusion criteria were the following: lack of informed consent; transfer from another hospital after >48 h of hospitalization; immunocompromised or HIV-positive status; pregnancy; palliative care; viral hepatitis; other chronic liver diseases (e.g., haemochromatosis, Wilson’s disease, toxic hepatitis, α1-antitrypsin deficiency, autoimmune liver disease); active malignancy; systemic autoimmune disease; excessive alcohol consumption (≥20 g/day in women, ≥30 g/day in men [48]); aspiration pneumonia; SARS-CoV-2 infection within the previous 90 days; active tuberculosis.

### 4.2. Data Collection and Definitions

We collected the following data: age, sex, comorbidities, chronic medications, presenting symptoms, day of illness on admission, treatment of CAP prior to admission and physical examination data on admission. Components of MetS were assessed according to International Diabetes Federation criteria—insulin resistance, abdominal obesity, hypertension and dyslipidemia [49]. Routine laboratory workup at admission and on the 5th day of hospitalization, including C-reactive protein (CRP), procalcitonin (PCT), lactate, electrolytes, glucose, urea, creatinine, bilirubin, aspartate aminotransferase (AST), alanine aminotransferase (ALT), gamma-glutamyl transferase (GGT), alkaline phosphatase (ALP), lactate dehydrogenase (LDH), troponin T, NT-pro-BNP, white-blood-cell count (WBC), with differential, hemoglobin, platelets, coagulation tests, arterial blood gases, IgM/IgA/IgG, total proteins, albumin, peripheral-blood lymphocyte immunophenotyping, serum-protein electrophoresis, urine biochemistry with sediment were collected, as well as a lipid panel (triglycerides, cholesterol, high-density lipoprotein [HDL], and low-density lipoprotein [LDL]). Additional serology tests (e.g., for systemic autoimmune disease and autoimmune liver disease) were performed when indicated. Routine microbiologic workup of all patients included blood and urine cultures with antimicrobial susceptibility testing, naso-/oropharyngeal swabs for SARS-CoV-2 and influenza A/B PCR, and urine antigen for *Legionella pneumophila*. Further microbiologic analyses included tracheal aspirate and BALF cultures with antimicrobial susceptibility, PCR tests on respiratory specimens (e.g., *Mycoplasma pneumoniae*, respiratory viruses), serology for *L. pneumophila* and other pathogens was performed at physicians’ discretion. All patients had a chest X-ray; CT scan was performed as indicated. In IMV patients, an additional BALF sample was collected for cytology, microbiology, biochemistry, and cytokine SEMA analyses.

Liver steatosis and fibrosis were evaluated within 72 h of admission using vibration-controlled transient elastography (VCTE, FibroScan^®^, Echosens, France) to obtain liver stiffness measurement (LSM) and the controlled attenuation parameter (VCTE-CAP). MASLD was diagnosed when VCTE-CAP indicated steatosis (<248 dB/m—no steatosis [S0], 248-267 dB/m—stage 1 steatosis [S1], 268-279 dB/m—stage 2 steatosis [S2], >279 dB/m—stage 3 steatosis [S3]) [50] and ≥1 cardiometabolic risk factor was present: BMI ≥ 25 kg/m^2^ or waist > 94 cm [men]/>80 cm [women]; fasting glucose ≥ 5.6 mmol/L, T2DM or antidiabetic therapy; blood pressure ≥ 130/85 mmHg or antihypertensive therapy; triglycerides ≥ 1.70 mmol/L or lipid-lowering therapy or HDL ≤ 1.0 mmol/L [men]/≤1.3 mmol/L [women], in the absence of other liver disease causes [48]. Other causes of liver disease were excluded in all patients by testing for hepatitis B virus (HBsAg), hepatitis C virus (anti-HCV) and HIV. As per predefined exclusion criteria, patients with significant alcohol consumption (>20 g/day for women, >30 g/day for men) were also excluded. Additional targeted laboratory tests (including autoimmune liver panels, ceruloplasmin, iron studies, α1-antitrypsin level) and abdominal imaging were performed when clinically indicated, to rule out autoimmune hepatitis, cholestatic liver diseases, Wilson’s disease, hemochromatosis, α1-antitrypsin deficiency, and drug-induced liver injury.

Disease severity scores computed from clinical and laboratory data included SIRS, SOFA, PSI, CURB-65, SMART-COP and the IDSA/ATS sCAP criterion count. Patients were reviewed daily until discharge; clinical course and outcomes were recorded. Clinical stabilization was considered the first day on which all of the following were present: temperature ≤ 37.8 °C; heart rate ≤ 100/min; respiratory rate ≤ 24/min; systolic BP ≥ 90 mmHg; SaO_2_ ≥ 90% or PaO_2_ ≥ 60 mmHg or SpO_2_ ≥ 94% without supplemental O_2_; ability to take oral/enteral nutrition; mental status similar to baseline [51].

### 4.3. Cytokine and Semaphorin Measurement

Additional serum samples were obtained within 48 h of admission and again 5 days later. For IMV patients, a BALF sample was collected within 24 h of endotracheal intubation. BAL was performed by conventional bronchoscopy or a non-bronchoscopic catheter (BAL Cath, Ballard Medical Products, Draper, UT, USA). Blood and BALF were stored at +4 °C and centrifuged within 48 h (blood 3500 rpm × 10 min; BAL 2500 rpm × 10 min). Supernatants were aliquoted and frozen at −80 °C pending analysis. Semaphorin (SEMA3A, SEMA3F, SEMA4D, SEMA5A, SEMA7A) concentrations were measured with commercial ELISA kits (ELISA kit, AssayGenie, Dublin, Ireland and for SEMA7A Abcam ELISA kit, Cambridge, UK). Serum concentrations of pre-selected cytokines/chemokines (predefined LEGENDplex panel: IL-4, IL-2, CXCL10 (IP-10), IL-1β, TNF-α (TNFSF2), CCL2 (MCP-1), IL-17A, IL-6, IL-10, IFN-γ, IL-12p70, TGF-β1 (Free Active), CXCL8 (IL-8)) were quantified using a flow-cytometer microsphere-based multiplex assay (LEGENDplex, Biolegend, San Diego, CA, USA), analyzed by flow cytometry BD FACSCanto II (Beckton Dickinson, Franklin Lakes, NJ, USA).

### 4.4. Statistical Analysis

A priori power analysis indicated that a sample size of at least 50 patients per group was required to detect statistically significant differences in semaphorin and cytokine concentrations between MASLD and non-MASLD groups across two time points with 80% power (α = 0.05). The observed group differences were generally larger than anticipated, and the resulting *p*-values and effect sizes suggest that the study was adequately powered not only for its primary immunological endpoints but also for selected clinical associations.

Data were presented descriptively as frequencies and medians with interquartile ranges (IQR). The cohort was stratified into MASLD and non-MASLD groups based on transient elastography findings, as previously described. Fisher’s exact test and the Mann–Whitney U test were used to compare categorical and continuous variables between groups. Pearson correlation analysis was performed to assess associations between semaphorin, cytokine, and chemokine concentrations and other laboratory parameters. To compare semaphorin and cytokine/chemokine levels between groups over time (Day 1 vs. Day 5), two-way repeated measures ANOVA was conducted, followed by post-hoc comparisons where appropriate. Delta (Δ) changes from Day 1 to Day 5 were derived from ANOVA outputs and used for graphical representation and interpretation of biomarker kinetics. These analyses were not adjusted for BMI or other metabolic syndrome components, as these traits are integral to the MASLD definition and may lie on the causal pathway between MASLD and immune alterations; adjusting for them could result in over-adjustment and underestimation of the total MASLD effect. In addition, the relatively small sample size within some subgroups limited the statistical power for multi-covariate adjustment in these analyses. Receiver operating characteristic (ROC) analysis was performed to determine optimal threshold values of biomarkers, and their prognostic value for pneumonia-related mortality was assessed using Kaplan–Meier survival analysis, with group comparisons made using the log-rank test. All statistical analyses were performed using GraphPad Prism version 10 (San Diego, CA, USA).

## 5. Conclusions

This prospective study identifies MASLD as a key immunometabolic driver of poor outcomes in sCAP, independently associated with early respiratory failure and shock. We reveal a novel MASLD-associated semaphorin–cytokine signature, characterized by early immune hyperactivation followed by possible immune exhaustion. These patterns were pathogen-specific and linked to clinical outcomes, with semaphorins SEMA7A and SEMA4D emerging as potential prognostic biomarkers. Our findings underscore the importance of the liver–lung axis in pneumonia and support incorporating MASLD into risk stratification frameworks, while highlighting semaphorin signaling as a promising target for future precision immunomodulatory strategies.

## Figures and Tables

**Figure 1 ijms-26-08095-f001:**
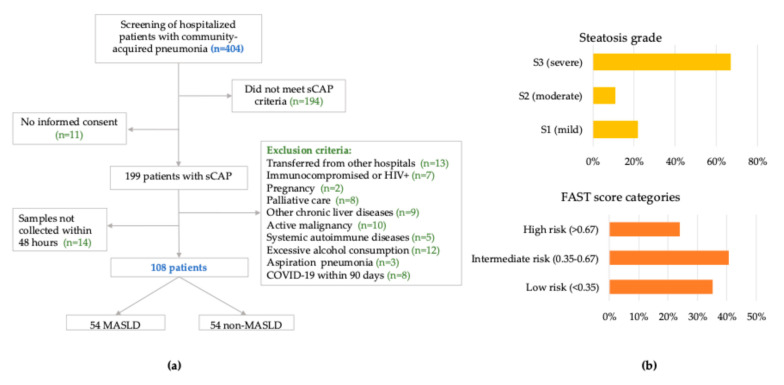
Study design flow chart and hepatic phenotype of MASLD patients. (**a**) Flow diagram showing patient screening, exclusions, and final allocation into MASLD and non-MASLD groups; (**b**) distribution of steatosis grades (by VCTE-CAP) and FAST score categories among MASLD patients at admission.

**Figure 2 ijms-26-08095-f002:**
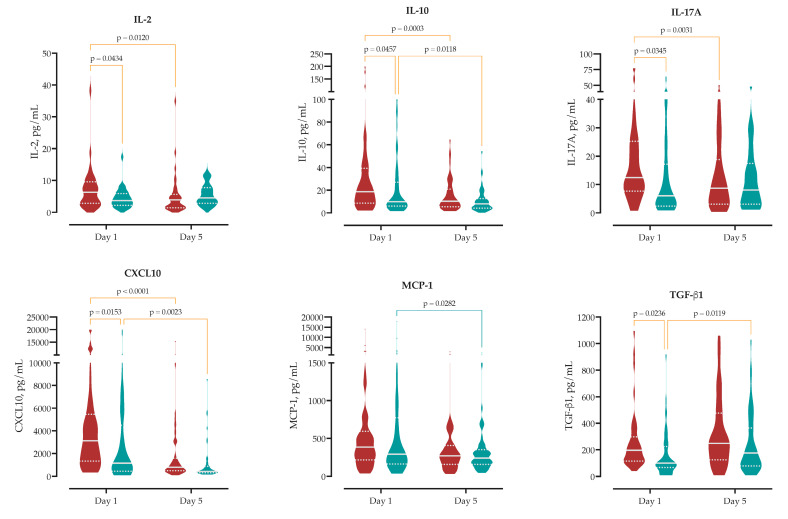
Temporal cytokine changes from Day 1 to Day 5 in patients with and without MASLD. Violin plots display cytokine dynamics over the early disease course (MASLD in red, non-MASLD in blue). Two-way ANOVA with multiple comparisons was used to assess time × MASLD interactions. Data are presented as median ± IQR, and adjusted *p*-values are reported for within-group and between-group comparisons. Full statistics are available in Appendix A.

**Figure 3 ijms-26-08095-f003:**
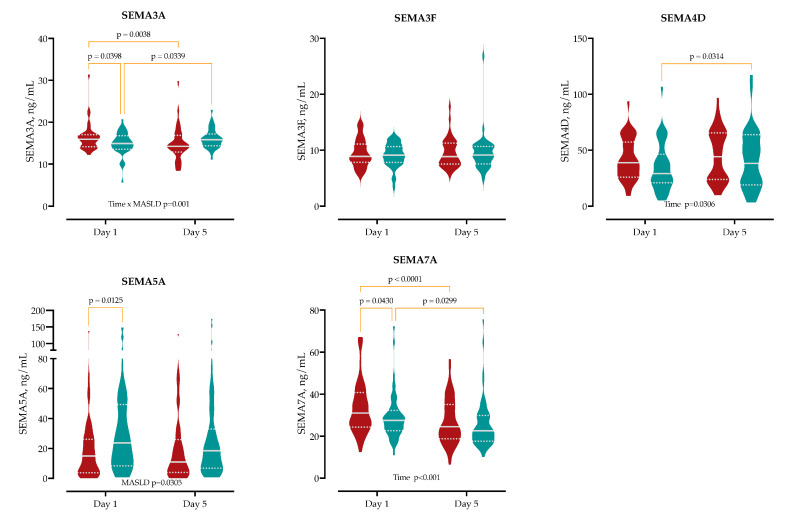
Serum semaphorin concentrations at admission (Day 1) and Day 5 in patients with and without MASLD. Violin plots show levels of SEMA3A, SEMA3F, SEMA4D, SEMA5A, and SEMA7A in MASLD (in red) and non-MASLD (in blue) patients across two time points. Statistical analysis was performed using two-way ANOVA with multiple comparisons. Significant between-group and within-group differences are indicated. Data are presented as medians with IQRs. Full statistics are provided in Appendix A.

**Figure 4 ijms-26-08095-f004:**
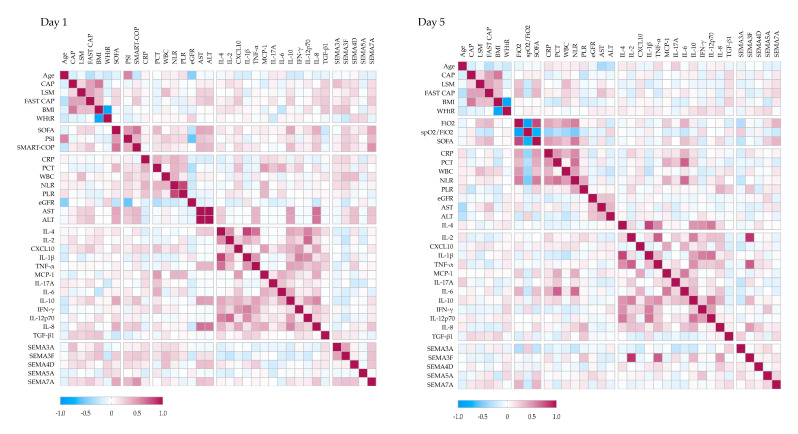
Correlation networks at hospital admission and on Day 5. Day 1 and Day 5 Pearson correlation matrices comprising clinical severity scores (SOFA, PSI, SMART-COP, etc.), liver disease metrics (VCTE-CAP, LSM, FAST), routine inflammatory markers, serum cytokines, and semaphorins. Cell color indicates the strength and direction of the correlation (blue r = −1, red r = +1).

**Figure 5 ijms-26-08095-f005:**
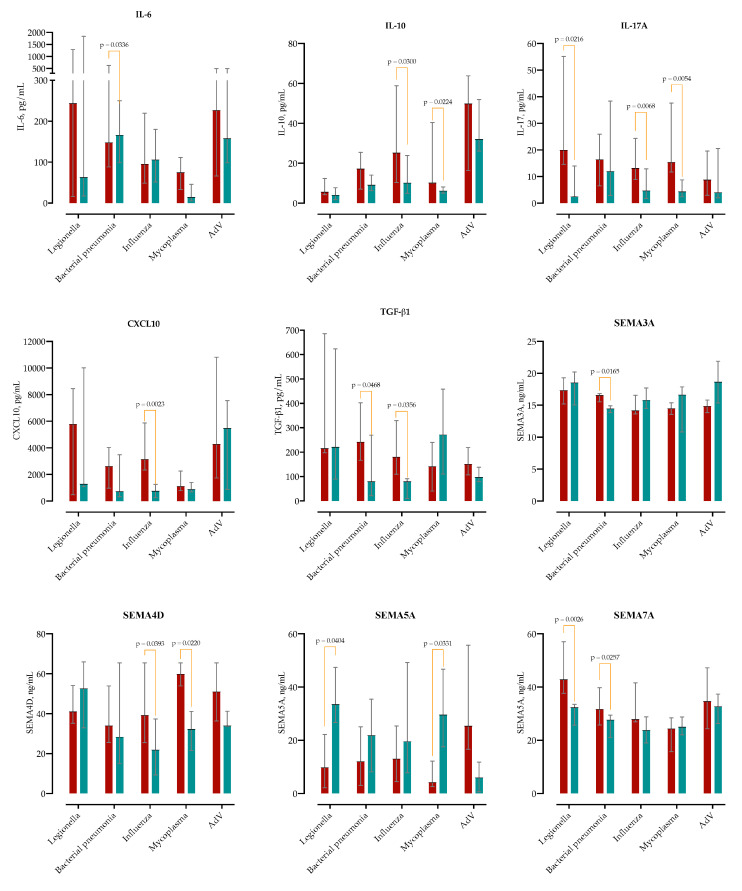
Admission cytokine concentrations stratified by MASLD status and microbial etiology. Medians ± IQR are shown for IL-6, IL-10, IL-17A, CXCL10, and TGF-β1 across five pathogen groups (MASLD in red and non-MASLD in blue). *p*-values represent between-group comparisons by two-way ANOVA with post-hoc correction.

**Figure 6 ijms-26-08095-f006:**
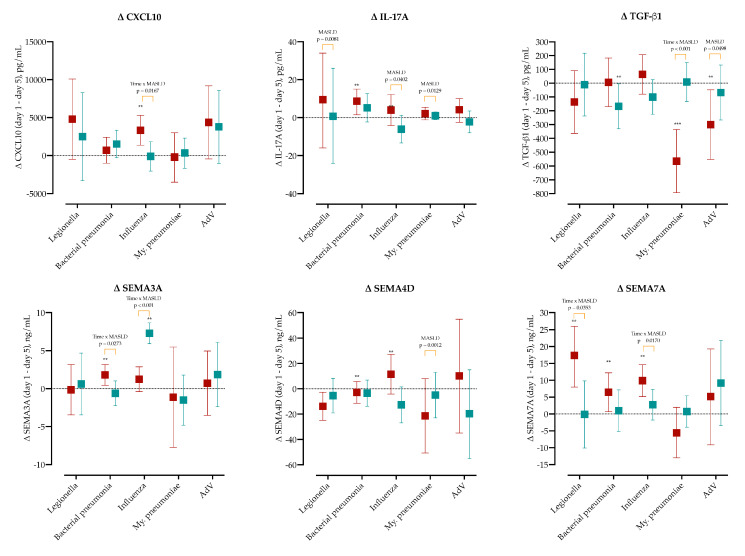
Dynamics of cytokines and semaphorins between Day 1 and Day 5, stratified by pathogen and MASLD status (MASLD in red, non-MASLD in blue). Shown are the changes (Δ = Day 5–Day 1) in CXCL10, IL-17A, TGF-β1, SEMA3A, SEMA4D, and SEMA7A across major etiological groups. Bars represent median change with 95% CI. Asterisks denote significant within-group changes ** *p* < 0.01. Orange brackets with *p*-values indicate significant between-group differences (MASLD vs non-MASLD) within the same pathogen class, based on two-way ANOVA with post-hoc test.

**Figure 7 ijms-26-08095-f007:**
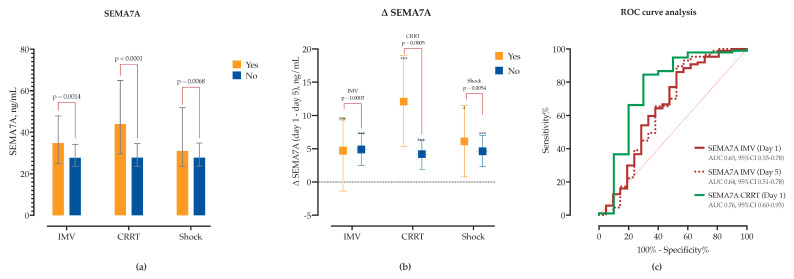
Association of SEMA7A with clinical outcomes in sCAP. (**a**) Baseline serum concentrations of SEMA7A stratified by the need for invasive mechanical ventilation (IMV), continuous renal replacement therapy (CRRT), and occurrence of septic shock. (**b**) Temporal change in SEMA7A levels from Day 1 to Day 5 within each outcome group. Asterisks indicate significant within-group differences (* *p* < 0.05, *** *p* < 0.001), and horizontal bars denote between-group comparisons with corresponding *p*-values from two-way ANOVA. (**c**) ROC curve analysis of SEMA7A levels on Day 1 and Day 5 as predictors of IMV and CRRT requirement. AUC values with 95% confidence intervals are indicated.

**Figure 8 ijms-26-08095-f008:**
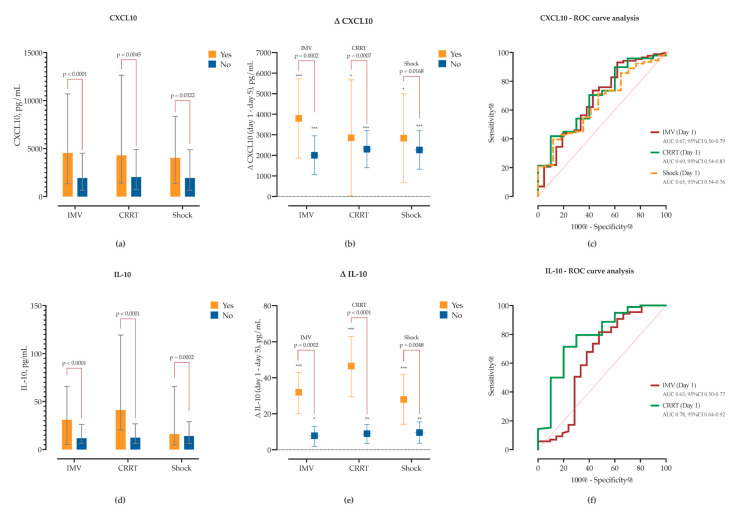
CXCL10 and IL-10 immune profiles and prognostic performance in sCAP. (**a**,**d**) Baseline serum concentrations of CXCL10 and IL-10 stratified by clinical outcomes: invasive mechanical ventilation (IMV), continuous renal replacement therapy (CRRT), and septic shock. (**b**,**e**) Temporal changes (Δ Day 5–Day 1) in CXCL10 and IL-10 levels for each outcome group. Asterisks indicate significant within-group changes (* *p* < 0.05, ** *p* < 0.01, *** *p* < 0.001); horizontal bars denote between-group comparisons with associated *p*-values from two-way ANOVA. (**c**,**f**) Receiver operating characteristic (ROC) curves for baseline CXCL10 and IL-10 levels as predictors of IMV, CRRT, and shock, with corresponding AUCs and 95% confidence intervals.

**Table 1 ijms-26-08095-t001:** Baseline patients’ characteristics.

	MASLD	Non-MASLD	*p*-Value
Age, years	58 (44–74)	63 (42–75)	0.7191
Male sex	40 (74.1%)	36 (66.7%)	0.5276
BMI (kg/m^2^)	32 (28–34)	26 (23–29)	<0.0001
Waist–hip ratio	1 (0.98–1.1)	0.98 (0.95–1)	0.0543
Waist–height ratio	0.61 (0.58–0.65)	0.53 (0.48–0.58)	<0.0001
Comorbidities			
Charlson comorbidity index	2 (0.75–4)	3 (0–6)	0.4172
Type 2 diabetes mellitus	11 (20.4%)	11 (20.4%)	1.0000
Arterial hypertension	32 (59.3%)	33 (61.1%)	0.8442
Chronic obstructive pulmonary disease	5 (9.3%)	4 (7.4%)	0.7277
Cardiovascular diseases	14 (25.9%)	15 (27.8)	0.8281
Chronic kidney disease	2 (3.7%)	7 (13.0%)	0.1610
Dyslipidemia	17 (31.5%)	15 (27.8%)	0.6734
Pneumonia severity at hospital admission			
Duration of symptoms before admission	6 (5–7)	7 (5–10)	0.0716
SOFA	3 (2–4.3)	3 (2–4)	0.7458
PSI	106 (84–121)	96 (71–131)	0.5373
CURB-65	2 (1–2)	2 (1–2)	0.2255
SMART-COP	4 (3–5)	4 (3–5)	0.4856
IDSA/ATS major criteria			
0	38 (70.4%)	45 (74.1%)	0.5044
1	10 (18.5%)	7 (13.0%)	
2	3 (5.6%)	5 (9.3%)	
3	3 (5.6%)	1 (1.9%)	
IDSA/ATS minor criteria	3 (3–4)	3 (3–4)	1.0000
ICU admission	14 (25.9%)	10 (18.5%)	0.4880
Non-ICU admission	40 (74.1%)	44 (81.5%)	

Data are presented as frequencies (%) or medians with IQRs. Fisher exact test or Mann–Whitney test were used to calculate *p*-values, as appropriate.

**Table 2 ijms-26-08095-t002:** Clinical course and outcomes of sCAP.

	MASLD	Non-MASLD	*p*-Value
ICU admission during hospitalization	18 (33%)	11 (20%)	0.1922
Advanced respiratory support	16 (29.6%)	7 (12.9%)	0.0586
Invasive mechanical ventilation	15 (27.8%)	6 (11.1%)	0.0287
Shock with vasopressors	13 (24.1%)	4 (7.4%)	0.0323
Continuous renal replacement therapy	7 (13%)	3 (5.6%)	0.1842
Time to clinical stabilization	7 (5–13)	6 (4–11)	0.6311
Duration of hospital stay	10 (6–16)	11 (7–14)	0.8824
In-hospital mortality	4 (7.4%)	1 (1.9%)	0.6362

Data are presented as frequencies (%) or medians with IQRs. Fisher exact test or Mann–Whitney test were used to calculate *p*-values, as appropriate.

## Data Availability

The datasets generated during and/or analyzed during the current study are available from the corresponding author upon reasonable request.

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
