# Peer review of "Metabolic Dysfunction-Associated Steatotic Liver Disease Shapes a Distinct Semaphorin–Cytokine Immune Signature in Severe Community-Acquired Pneumonia"

_ijms, 2025, doi:10.3390/ijms26168095_

Round 1
Reviewer 1 Report
Comments and Suggestions for Authors
The authors conducted extensive comparative and correlation analyses, providing a comprehensive understanding of clinical parameters in patients with and without MASLD. The study identified MASLD as a key immunometabolic driver of poor outcomes in sCAP and highlighted the significance of the liver-lung axis. Additionally, two potential prognostic biomarkers—SEMA7A and SEMA4D—were identified. The results are clearly presented. However, validation of these findings in an independent cohort would enhance the robustness and generalizability of the conclusions.
Author Response
The authors conducted extensive comparative and correlation analyses, providing a comprehensive understanding of clinical parameters in patients with and without MASLD. The study identified MASLD as a key immunometabolic driver of poor outcomes in sCAP and highlighted the significance of the liver-lung axis. Additionally, two potential prognostic biomarkers—SEMA7A and SEMA4D—were identified. The results are clearly presented. However, validation of these findings in an independent cohort would enhance the robustness and generalizability of the conclusions.
Authors' response: We thank the reviewer for the positive assessment of our work and the constructive suggestion. We fully agree that validation in an independent cohort would further strengthen our conclusions. While such validation is beyond the scope of the present study, we have emphasized this point in the Discussion as a key direction for future research, and we are currently planning follow-up studies to address it.
Reviewer 2 Report
Comments and Suggestions for Authors
This paper presents a new and interesting data contributing to unveil new relationships between potencial biomarkers in diseases poorly approached on peer reviewed literature.
It is a well designed methodology regarding the main objectives, even though it has some limitations, as acknowledge by the authors, however, it extracts very interesting and applicable data, mainly for clinicians, to pursue new therapeutic approaches to interrelated diseases considering the under or overexpression of certain biomarkers easily determined by a simple serum sample.
Although, I would make some suggestions on the data presentation, namely the resolution of graphics on figures 2 to 8, because, even on pdf version, when zooming in, letters and numbers became blur. In supplementary data the graphics are absolutely legible but, in the main paper version, that is not the case.
In conclusion, I really think this is a very pertinent approach to precision medicine and I would like to see further studies exploring this thematic.
Author Response
This paper presents a new and interesting data contributing to unveil new relationships between potencial biomarkers in diseases poorly approached on peer reviewed literature.
It is a well designed methodology regarding the main objectives, even though it has some limitations, as acknowledge by the authors, however, it extracts very interesting and applicable data, mainly for clinicians, to pursue new therapeutic approaches to interrelated diseases considering the under or overexpression of certain biomarkers easily determined by a simple serum sample.
Although, I would make some suggestions on the data presentation, namely the resolution of graphics on figures 2 to 8, because, even on pdf version, when zooming in, letters and numbers became blur. In supplementary data the graphics are absolutely legible but, in the main paper version, that is not the case.
In conclusion, I really think this is a very pertinent approach to precision medicine and I would like to see further studies exploring this thematic.
Authors' response: We thank the reviewer for the positive and encouraging comments regarding the novelty, clinical relevance, and methodological design of our study. We have replaced all figures in the main manuscript with higher-resolution versions.
Reviewer 3 Report
Comments and Suggestions for Authors
The study focuses on the impact of metabolic dysfunction-associated steatotic liver disease (MASLD) on the immune response in severe community-acquired pneumonia (sCAP), particularly the semaphorin-cytokine signaling axis. A prospective cohort study design was employed, enrolling 108 sCAP patients with balanced grouping. Pathogen-stratified analyseswere conducted, accounting for the modifying effects of pathogens on immune responses, making the findings more targeted.
1.All images need to be replaced with high-resolution versions.
2.The article mentions the use of VCTE-CAP and FAST scores for diagnosing MASLD but does not specify the cut-off values or clearly state whether other liver diseases were excluded; the specific diagnostic criteria for MASLD should be added in the methods section, along with the basis for excluding other liver diseases, to enhance the reproducibility of the results.
3.Pathogen-stratified analysis was performed in the article, but the sample sizes of each subgroup were not reported, which may lead to insufficient statistical power for subgroup analysis. The sample sizes of each pathogen subgroup should be added in the results section, and the statistical power of the subgroup analysis should be explained through power analysis, or the impact of small sample sizes on the results should be discussed.
4.The article reports extremely low in-hospital mortality, while the ICU mortality for sCAP is usually 20%-30%. This may be due to insufficient follow-up time or milder patient conditions. The reasons for the low mortality should be explained in the discussion section, and 30-day mortality data should be added if available; meanwhile, the interpretation of the mortality results should be adjusted.
5.The article mentions “insufficient immune repair” in MASLD patients but does not deeply explain the interaction mechanism between semaphorins and cytokines, nor does it explore the root cause of immune dysregulation in combination with the metabolic abnormalities of MASLD. The discussion section should strengthen the discussion on mechanisms, explain the mechanism underlying the immune characteristics of MASLD patients combined with existing studies, and meanwhile link to metabolic abnormalities to illustrate the specific role of the “liver-lung axis”.
6. The BMI of MASLD patients was significantly higher than that of the non-MASLD group (32.1 vs 25.6), and BMI itself may affect immune responses, but it is not clear whether BMI was adjusted for in the multivariate analysis in the article. BALF was only obtained from intubated patients, which cannot represent the local immune responses of all sCAP patients and may lead to resultant bias.
7. The limitations of the BALF samples should be explained in the discussion section, and it is suggested that future studies include BALF samples from non-intubated patients to compare the differences between local and systemic immune responses.
Author Response
The study focuses on the impact of metabolic dysfunction-associated steatotic liver disease (MASLD) on the immune response in severe community-acquired pneumonia (sCAP), particularly the semaphorin-cytokine signaling axis. A prospective cohort study design was employed, enrolling 108 sCAP patients with balanced grouping. Pathogen-stratified analyseswere conducted, accounting for the modifying effects of pathogens on immune responses, making the findings more targeted.
1. All images need to be replaced with high-resolution versions.
Authors' response: We have replaced Figures 2–8 in the main manuscript with higher-resolution versions.
2. The article mentions the use of VCTE-CAP and FAST scores for diagnosing MASLD but does not specify the cut-off values or clearly state whether other liver diseases were excluded; the specific diagnostic criteria for MASLD should be added in the methods section, along with the basis for excluding other liver diseases, to enhance the reproducibility of the results.
Authors' response: We thank the reviewer for this valuable suggestion. The cut-off values for VCTE-CAP and the criteria for MASLD diagnosis, including the requirement for ≥1 cardiometabolic risk factor, were already stated in the Methods section of the original submission. The exclusion of other liver diseases was also mentioned; however, to improve clarity and reproducibility, we have now expanded this part to specify that all patients were tested for hepatitis B virus (HBsAg), hepatitis C virus (anti-HCV), and HIV, and that additional targeted testing (autoimmune liver panels, ceruloplasmin, iron studies, α1-antitrypsin level) and abdominal imaging were performed when clinically indicated. We also now explicitly stated that predefined exclusion criteria included significant alcohol consumption (>20 g/day for women, >30 g/day for men). The revised text is provided in the Methods section (page 19-20, line 699-706) of the revised manuscript.
3. Pathogen-stratified analysis was performed in the article, but the sample sizes of each subgroup were not reported, which may lead to insufficient statistical power for subgroup analysis. The sample sizes of each pathogen subgroup should be added in the results section, and the statistical power of the subgroup analysis should be explained through power analysis, or the impact of small sample sizes on the results should be discussed.
Authors' response: We thank the reviewer for this important observation. In the revised manuscript, we have now reported the sample sizes for each pathogen subgroup in the Results section (see page 4, line 135-138, and Supplementary Figure S1). Given the exploratory nature of the pathogen-stratified immune analysis and the relatively small sample sizes in some subgroups, we did not perform a priori power calculations for these comparisons. We have now explicitly acknowledged in the Discussion that some subgroup analyses may be underpowered, and therefore these findings should be interpreted with caution and considered hypothesis-generating.
4. The article reports extremely low in-hospital mortality, while the ICU mortality for sCAP is usually 20%-30%. This may be due to insufficient follow-up time or milder patient conditions. The reasons for the low mortality should be explained in the discussion section, and 30-day mortality data should be added if available; meanwhile, the interpretation of the mortality results should be adjusted.
Authors' response: We thank the reviewer for this important observation. In our cohort, the overall in-hospital mortality was indeed lower than previously reported for ICU-treated sCAP. This is likely related to the fact that our study included both ICU and non-ICU hospitalized patients with sCAP, with the majority (77%) managed outside the ICU, as well as the high proportion of patients with relatively preserved baseline functional status. We have now added available 30-day mortality data to the Results (3.7%). In the Discussion, we have clarified that the lower mortality rate may reflect differences in patient selection, case mix, and the setting of care, and we have adjusted our interpretation accordingly (Page 18, line 608-614).
5. The article mentions “insufficient immune repair” in MASLD patients but does not deeply explain the interaction mechanism between semaphorins and cytokines, nor does it explore the root cause of immune dysregulation in combination with the metabolic abnormalities of MASLD. The discussion section should strengthen the discussion on mechanisms, explain the mechanism underlying the immune characteristics of MASLD patients combined with existing studies, and meanwhile link to metabolic abnormalities to illustrate the specific role of the “liver-lung axis”.
Authors' response: We thank the reviewer for this comment. In the revised manuscript, we have expanded the Discussion to provide a more detailed mechanistic interpretation of our findings (Page 17-18, line 588-600). Specifically, we have integrated evidence from prior studies on semaphorin signaling, cytokine regulation, and metabolic dysfunction to explain how MASLD-associated metabolic abnormalities—such as insulin resistance, chronic low-grade inflammation, and altered hepatic lipid metabolism—may impair both early inflammatory containment and subsequent tissue repair in sCAP. We also elaborate on how specific semaphorins (e.g., SEMA7A, SEMA4D) interact with cytokine networks (e.g., IL-10, TGF-β1, CXCL10) to modulate leukocyte recruitment, macrophage polarization, and fibroproliferative responses, and how these processes may be dysregulated in the setting of MASLD.
6. The BMI of MASLD patients was significantly higher than that of the non-MASLD group (32.1 vs 25.6), and BMI itself may affect immune responses, but it is not clear whether BMI was adjusted for in the multivariate analysis in the article. BALF was only obtained from intubated patients, which cannot represent the local immune responses of all sCAP patients and may lead to resultant bias.
Authors' response: We thank the reviewer for this important observation. The cytokine and semaphorin analyses in our study were exploratory and performed using two-way ANOVA to examine the effects of MASLD status and time. We did not adjust for BMI or other metabolic syndrome components for two main reasons: (a) conceptually, these traits are integral to the MASLD definition and likely lie on the causal pathway between MASLD and immune alterations; adjusting for them could constitute over-adjustment and underestimate the total MASLD effect; and (b) practically, the relatively small sample sizes within certain subgroups limited statistical power for multi-covariate adjustment in these analyses. We have now clarified this in the Statistical Analysis section (Page 20, line 749-754) and acknowledged in the Discussion (Page 18, line 606-608) that BMI and other metabolic syndrome traits may contribute to the observed immune differences.
7. The limitations of the BALF samples should be explained in the discussion section, and it is suggested that future studies include BALF samples from non-intubated patients to compare the differences between local and systemic immune responses.
Authors' response: We agree with the reviewer. We have expanded the BALF-related limitation in the Discussion to explicitly note that our sampling was limited to intubated patients and may not reflect the local immune responses in non-intubated sCAP patients. We have also added that future studies should aim to include BALF or alternative lower respiratory tract sampling from both intubated and non-intubated patients to enable direct comparison of local and systemic immune responses (Page 18, line 614-619).
Round 2
Reviewer 3 Report
Comments and Suggestions for Authors
No other suggestions